# Generation of Immune Modulating Small Metabolites—Metabokines—By Adult Schistosomes

**DOI:** 10.3390/pathogens14060526

**Published:** 2025-05-24

**Authors:** Patrick J. Skelly, Akram A. Da’dara

**Affiliations:** Molecular Helminthology Laboratory, Department of Infectious Disease and Global Health, Cummings School of Veterinary Medicine, Tufts University, North Grafton, MA 01536, USA; akram.da_darah@tufts.edu

**Keywords:** parasite, blood fluke, Schistosoma, immunomodulation, metabolomics

## Abstract

Schistosomes are intravascular parasitic worms that cause the debilitating tropical disease schistosomiasis, affecting >200 million people worldwide. How the worms survive within the body of immunocompetent hosts for many years is unclear. Here, using chromatography and mass spectrometry, we report on the ex vivo ability of adult *Schistosoma mansoni* worms to modulate the levels of 27 small molecule (often immunomodulatory) metabokines in murine plasma. Schistosomes significantly alter the relative amounts of most (16) of these molecules. Three (inosine, genistein, and glucose) are significantly decreased in the presence of the parasites. While levels of several immunomodulatory metabolites from the kynurenine pathway (kynurenine, kynurenic acid, and xanthurenic acid) remain unchanged, levels of anthranilate (an endogenous regulator of innate immunity) are significantly increased. Of particular interest are increases in levels of metabolites that are known to skew immune responses in a manner that is seen following natural schistosome infection, such as by promoting Th2 immunity (succinate), Treg generation (lactate) and M2 macrophage polarization (lactate and succinate). In addition, significant increases are also observed for 2-hydroxyglutarate, adenine, hypoxanthine, xanthine, myoinositol, betaine and N-acetylglucosamine. Each of these compounds can have immunosuppressive effects that could impact host immunological status and contribute to schistosome survival.

## 1. Introduction

Schistosomes are parasitic flatworms responsible for schistosomiasis, a neglected tropical disease affecting over 200 million people worldwide. The schistosome life cycle begins when eggs released in urine or feces reach freshwater, where they hatch into free-swimming larval forms called miracidia. These infect specific freshwater snails, their intermediate hosts, and develop into sac-like structures called sporocysts, which, through asexual replication, produce new larval forms called cercariae. The cercariae emerge from the snails and actively penetrate the skin of their definitive (human) hosts upon contact with contaminated water. Inside, they transform into juvenile forms called schistosomula and these migrate through the bloodstream to eventually mature into adult worms in the mesenteric veins, or bladder vasculature, depending on the species. Male and female worms pair and females produce eggs, which may be excreted to continue the cycle or, if they become trapped in the internal tissues, can cause chronic inflammation and organ damage. Schistosomiasis can lead to severe complications such as liver fibrosis, bladder cancer, and cognitive impairment [1,2].

Schistosomes can survive within the body of an immunocompetent host for many years [2]. How they avoid being eliminated by host immune effectors is a matter of much research. The worms produce a suite of molecules that are known (or hypothesized) to modulate host immunity, negating its damaging effects and promoting their survival [3]. Some of these molecules are schistosome ectoenzymes that are exposed at the parasite surface where they interact directly with host biochemistry [4]. In addition, the worms produce a plethora of excretions/secretions (ES) that can impact host immunity [5]. Here, we expand the list of potential schistosome immunomodulators with an examination of a group of previously overlooked small molecules, now collectively called metabokines, that are known to profoundly influence immunological processes [6]. A “metabokine” is defined as a small, naturally produced chemical derived from cellular metabolism that can act as a signaling molecule, locally or systemically, to regulate physiology and biochemistry. The identity and influence of some metabokines has been surprising. For example, lactate, which has been traditionally viewed solely as a metabolic byproduct of anaerobic glycolysis, has emerged in recent years as a potent metabokine [7]. Lactate can exert immunomodulatory impact in two ways: First, it can be directly transported into immune cells through their monocarboxylate transporters (MCTs) [8] where it can reprogram, e.g., macrophage metabolism, to inhibit expression of pro-inflammatory factors [9]. Second, extracellular lactate can act directly as a signaling molecule by binding to cell surface lactate receptors such as G-protein-coupled receptor (GPR) 81 and GPR132 to drive cellular responses [10]. Similarly, several intermediates of the tricarboxylic acid (TCA) cycle, such as succinate, citrate and fumarate have now, unexpectedly, been shown to play significant roles in modulating immune responses by influencing both innate and adaptive immune cells [11,12,13]. Additionally, metabolites from the kynurenine pathway of tryptophan metabolism, including kynurenine, kynurenic acid, and anthranilate, can play crucial roles in modulating immune responses by influencing immune cell activation and cytokine production [14]. Here, we report on the ability of adult *Schistosoma mansoni*, when incubated in murine plasma ex vivo, to impact the levels of these, and other, metabokines over time.

## 2. Materials and Methods

### 2.1. Parasites and Mice

In this work, the Puerto Rican strain (NMRI) of *Schistosoma mansoni* was used. *Biomphalaria glabrata* snails, infected with *S. mansoni* were obtained from the NIAID Schistosomiasis Resource Center of the Biomedical Research Institute (BRI), Rockville, MD, USA, and maintained in our laboratory. Cercariae (infectious larvae) were prepared by exposing infected snails to light for 1–2 h to induce shedding. Next, Swiss Webster mice were infected with ~100 cercariae each and, seven weeks later, adult male and female parasites were recovered by vascular perfusion [15]. Parasites were cultured for 48 h in DMEM/F12 medium supplemented with 10% heat-inactivated fetal bovine serum, 200 µg/mL streptomycin, 200 U/mL penicillin, 1 µM serotonin, 0.2 µM Triiodo-l-thyronine, 8 µg/mL human insulin and were maintained at 37 °C, in an atmosphere of 5% CO_2_ [16]. All protocols involving animals were approved by the Institutional Animal Care and Use Committee (IACUC) of Tufts University.

### 2.2. Metabokine Detection in Murine Plasma

Blood was collected into heparinized tubes from the tail veins of 10 uninfected Swiss Webster mice. Blood cells were pelleted by brief centrifugation and the plasma generated was pooled and aliquoted. Adult schistosomes (approximately 50 pairs) were incubated in one 500 µL murine plasma aliquot which was incubated at 37 °C. A control aliquot (without worms) was similarly incubated at 37 °C. Samples, collected at baseline (0 min) and after 20 and 60 min incubation with or without parasites, were subjected to metabolomic analysis at Metabolon Inc. Three to four samples per treatment/time point were tested. Briefly, each plasma sample was prepared by solvent extraction, and the resulting extract was split into equal parts and then applied to gas chromatography/mass spectrometry (GC/MS) and liquid chromatography tandem MS (LC/MS/MS) platforms [17]. Metabokines were each identified by their retention time and mass by comparison to purified standards. The raw data values for each biochemical, provided in Appendix A, were normalized and rescaled and results are presented relative to the baseline measurement (0 min), set at 1.

### 2.3. Statistical Analysis

Data are presented as box plots, displaying the median and mean values for each sample where each box bounds the upper and lower quartile, with error bars indicating the maximum (upper) and minimum (lower) distribution. One-way analysis of variance (ANOVA) with Tukey post hoc analysis was used to compare the means between the different normalized groups using GraphPad Prism 10.4. *p* values less than 0.05 were considered significant.

## 3. Results

When adult schistosomes are incubated in murine plasma, they rapidly alter its composition. For instance, Figure 1 depicts changes in levels of glucose (A) and lactate (B), measured in plasma containing adult worms (+), or in control plasma lacking worms (−), at the indicated time points. Significant differences between levels of a metabolite in plasma containing worms versus the equivalent control plasma lacking worms are indicated.

As depicted in Figure 1A, the level of glucose significantly decreases over time, and the decrease detected at 20 min post incubation becomes even greater at the 60 min time point. Simultaneously, as shown in Figure 1B, levels of lactate increase significantly in like manner.

Figure 2 depicts levels of selected tricarboxylic acid (TCA) cycle intermediates in plasma containing adult schistosomes (+) or in control plasma lacking schistosomes (−) at the indicated time points. (A) is a representation of the TCA cycle. Note that not all TCA cycle biomolecules were contained within the metabolite panel; only those bounded by orange boxes in Figure 2A were measured. Other panels depict levels of these metabolites: citrate (B), aconitate (C), α-ketoglutarate (D), succinate (E), fumarate (F), malate (G) and the α-ketoglutarate derivative, 2-hydroxyglutarate (H), over time. Most TCA cycle intermediates show no significant change in level over time, including citrate (panel B), α-ketoglutarate (D), fumarate (F) and malate (G). The most striking change is seen in the level of succinate (E) at both 20 and 60 min; there is also a significant increase in the level of aconitate (C), but only at the 60 min time point. Finally, there is a significant increase in the level of the α-ketoglutarate derivative, 2-hydorxyglutarate (H), at both time periods.

Several metabolites of the kynurenine pathway are powerful metabokines [18] and Figure 3 depicts changes in levels of these biomolecules in plasma containing adult schistosomes (+) or in control plasma lacking schistosomes (−) at the indicated time points. (A) is a representation of part of the kynurenine pathway and the remaining panels depict levels of the following pathway components (bounded by orange boxes in A): tryptophan (B), kynurenine (C), kynurenate (D), anthranilate (E) and xanthurenate (F), over time. No significant differences were recorded for the levels of most kynurenine pathway metabolites in plasma containing worms vs. equivalent, control plasma lacking worms. This is true for tryptophan (B), kynurenine (C), kynurenate (D) and xanthurenate (F). However, significant differences are recorded for anthranilate at both the 20 and 60 min time points (Figure 3E).

Purinergic metabolites can have profound impacts on immune cell function and Figure 4 depicts changes in the levels of selected purinergic biomolecules in plasma containing worms (+) or in control plasma lacking worms (−) at the indicated time points. (A) represents some steps in purine metabolism; the remaining panels depict levels of the following pathway components (bounded by orange boxes in A): adenine (B), inosine (C), hypoxanthine (D) and xanthine (E) in plasma containing adult schistosomes (+) or in control plasma lacking schistosomes (−), at the indicated time points. Levels of all four measured metabolites change significantly at all time points. Adenine levels increase substantially, to about 10 times their basal level (panel B). Both hypoxanthine (D) and xanthine (E) levels also increase, but to a lesser (relative) extent compared to adenine. In contrast, inosine levels decline naturally (in plasma lacking schistosomes) but schistosomes significantly speed up this decline.

Figure 5 depicts changes in levels of several miscellaneous metabolites of diverse biochemical classes that have been shown to impact host immunology. Some of these show striking increases in plasma containing adult schistosomes (+) compared to control plasma lacking schistosomes (−), including myo-inositol (A), betaine (B) and N-acetylglucosamine (C). Others show more modest increases, such as gentisate (D), or declines, such as genistein (E). Some show a modest increase (such as sorbitol, F) or decrease (such as azelate, G) just at the 60 min time point. Some metabolites show no significant (ns) change over time, such as fructose (H). Observations regarding glycerol (I) differ from the rest in that its level increases modestly in control plasma and schistosomes prevent this increase; they maintain glycerol levels a little below baseline (the 0-time measurement). Significant differences between levels of a metabolite in plasma containing worms vs. equivalent, control plasma lacking worms are indicated.

Table 1 summarizes the changes in metabokine levels reported here; the raw data for all chemicals monitored are presented in Appendix A, along with information concerning each chemical. In sum, we find that adult schistosomes significantly altered the relative amounts of more than half of the analyzed plasma metabokines (16/27, as listed in Table 1). The remainder (11/27) exhibited no significant changes in level. Of those that changed, the levels of the majority were significantly increased (13/16), while only 3 metabolites were significantly decreased in the presence of the parasites (gray boxes in Table 1).

## 4. Discussion

Schistosomes are long-lived parasites that inhabit the bloodstreams of their hosts. In this work, we report on how these parasites radically alter the murine plasma metabolome, with especial emphasis on changes in small metabolites, now called metabokines, some of which can dramatically impact host immunological functions.

Here, we first demonstrate that, as expected, glucose levels rapidly decrease in plasma in which worms are incubated. It is known that intravascular schistosomes take in glucose in great quantities from host blood directly across their body surface via sugar transporter proteins (SGTPs) located in the worm’s outer tegumental membranes [19,20]. Imported glucose is metabolized largely by glycolysis whose final metabolic product in the worms is lactate [21]. Adult schistosomes are considered homolactate fermenters [22]. Lactate can be excreted across the parasite’s tegument via aquaporin proteins that, like the SGTPs just mentioned, are also located in the host-interactive surface membranes [23]. Schistosomes ex vivo are known to rapidly acidify their culture medium by excreting large quantities of lactate [22]. In agreement with this observation, we show here that, in murine plasma that contains adult worms ex vivo, lactate levels rise significantly, shortly after the worms are incubated in the plasma, and levels continue to go up with more time in culture. For years, lactate was considered a waste product of cellular metabolism, but now it has emerged as a potent metabokine, exerting a wide range of immunomodulatory effects [10,24,25]. For instance, extracellular lactate has been shown to suppress dendritic cell differentiation and maturation [26], to suppress T cell proliferation and cytokine production [27,28] and to regulate the functions of mast cells [29]. The extraordinarily wide range of effects that lactate can exert on these, and other, immune cells is reviewed in [26,30,31]. Perhaps of greater relevance in the context of schistosome infection is the ability of lactate to drive alternatively activated macrophage (M2) differentiation, since this is a prominent immunological feature seen following schistosome infection [32]. Lactate has been shown to activate ERK/STAT3 signaling [33] and to drive increased expression of Vascular Endothelial Growth Factor (VEGF) and arginase (ARG1) in macrophages, indicative of their differentiation into the M2-like phenotype [34,35]_._ Lactate has similarly been shown to drive T regulatory (Treg) cell polarization from naïve T cells, inducing NF-kB activity and Foxp3 expression [28]. Increased Treg proliferation and activation is another immunological feature that is characteristic of schistosome infection [36]. Therefore, among other effects, the prodigious production and release of lactate by schistosomes may play some role in driving host immunity towards a more tolerant state by promoting Treg and M2 macrophage generation.

### 4.1. TCA Cycle Intermediates

The tricarboxylic acid (TCA), or Kreb’s, cycle is a series of biochemical reactions that release stored energy in the form of ATP and reducing equivalents through the oxidation of acetyl-CoA derived from fats, carbohydrates and protein. While, as described above, intravascular schistosomes favor glycolysis, they do possess active mitochondria and can also produce energy through aerobic metabolism [37]. Thus, they produce TCA cycle intermediates; several of these have been shown, as for lactate, to exert considerable impacts on immune effector functions. For instance, depending on its concentration, citrate can either augment or inhibit pro-inflammatory cytokine production via modulation of inflammatory gene activation in human acute monocytic leukemia cell line (THP-1) cells [12]. Exogenous alpha-ketoglutarate (αKG) can impair the differentiation and maturation of monocyte-derived dendritic cells (moDCs) that have been induced with lipopolysaccharide (LPS)/interferon (IFN)-γ and can decrease their capacity to induce Th1 cells [38]. Further, when added to the diet of piglets, αKG can alleviate intestinal inflammation induced by lipopolysaccharide (LPS) through the Th17/Treg immune response signaling pathway [39]. A fumarate derivative, dimethyl fumarate, is used therapeutically and has been shown to shift the immune environment toward an anti-inflammatory cell profile, promoting regulatory B and T cells and reducing inflammatory cytokines in T and B cells [40]. Dimethyl fumarate treatment shifts the immune environment toward an anti-inflammatory cell profile, while maintaining protective humoral immunity [13]. Monomethyl fumarate has been shown to induce a Th1 to Th2 lymphocyte shift, to inhibit dendritic cell maturation and to impinge on other immune cell lineages (reviewed in [41]). While these TCA cycle intermediates can clearly act as metabokines, we detect no significant change in plasma levels of either citrate, alpha-ketoglutarate, fumarate or malate in plasma containing worms compared to control plasma. In contrast, there is a substantial and striking increase in another TCA cycle intermediate—succinate—in plasma containing adult schistosomes. At both the 20 and 60 min incubation timepoints, significantly greater amounts of succinate can be detected in plasma containing worms compared to control plasma lacking worms. Succinate is among the better studied metabokines [42]. In the extracellular environment, succinate can exert substantial physiological effects by activating the G protein-coupled receptor SUCNR1 (succinate receptor 1, also known as GPR91) [43]. By engaging SUCNR1 on immune cells, succinate can modulate their pro-inflammatory or anti-inflammatory activities. For instance, succinate binds to SUCNR1 on tuft cells, stimulating them to release IL-25; this acts on class 2 innate lymphoid cells (ILC2s) to promote the secretion of IL-13, which directly enhances type 2 immunity [44]. IL13 also acts on dendritic cells (DCs) to promote their migration to the mesenteric lymph nodes; this induces CD4+ T cell polarization into Th2 cells, also promoting overall type 2 immunity [45]. Since chronic schistosomiasis is marked by Th2 immunity [46,47], our data suggest that schistosome secretion of metabokines like succinate could contribute to the polarized immune response that is characteristic of this infection. Further, like lactate, as noted earlier, succinate can also drive macrophage polarization; it has been reported that culturing peritoneal murine macrophages in the presence of succinate results not only in the upregulation of the M2-related genes *Arg1, TGF-β* and *CD206*, but also in a morphological transition from “spherical to shuttle-shaped cells with increased tentacles” [48]. Similarly, succinate secreted by cancer cells is reported to activate SUCNR1 signaling, converting murine peritoneal macrophages into the M2-polarized form [49]. Additional effects of succinate on these and other cells of the immune system are reviewed in [11,45,50].

The mean level of another metabolite of the tricarboxylic acid (TCA) cycle, aconitate, also goes up in the presence of worms, but significantly so only at the 60 min time point. While aconitate has not been extensively studied as an immunomodulator, its derivative, itaconate, can exert profound anti-inflammatory effects by inhibiting the NLRP3 inflammasome, reducing the production of pro-inflammatory cytokines such as IL-1β and IL-18 [51]. Additionally, itaconate can modulate macrophage activation and has been implicated in the regulation of reactive oxygen species (ROS) production [52].

Finally, as noted above, we see no significant changes in the levels of αKG in plasma containing worms versus control plasma at any time point examined here. However, we do note significant accumulation in plasma containing schistosomes of a related immunomodulatory metabolite—2-hydroxyglutarate (2-HG, which contains a hydroxyl group rather than a ketone group in the second carbon position). At both the 20 min time point and at the 60 min time point, the level of 2-HG is significantly raised in plasma containing worms versus control plasma. 2-HG can be taken into cells through dicarboxylate transporters, such as solute carrier family 13 member 3 (SLC13A3) [48]. Regarding immunomodulation, treatment of mouse or human macrophages with 2-HG suppresses LPS-induced secretion of key inflammatory mediators such as IL-6, TNF-α and nitric oxide (NO), and can reduce expression of inflammatory markers like CD40 and inducible nitric oxide synthase (iNOS) [53]. In addition, exogenously administered 2-HG can impact T cell function; by impairing the ability of CD8+ T cells to degranulate upon antigen restimulation, 2-HG makes the cells poorly cytotoxic [54]. Finally, 2-HG can inhibit CD8+ T cell proliferation in a dose-dependent manner [55].

### 4.2. The Kynurenine Pathway

The kynurenine pathway, one way in which tryptophan is metabolized, triggers the production of a variety of metabolites, such as kynurenine, kynurenic acid and anthranilic acid, and these metabolites are reported to contribute to immune tolerance [14,56]. For instance, kynurenine can suppress cytotoxic T cell function [57] and can skew the differentiation of naïve T cells to Treg cells rather than Th17 cells in culture [58]. While we detect no significant changes in kynurenine or kynurenate levels or in the levels of another metabolite of this pathway—xanthurenate—in plasma containing schistosomes, there is a striking and significant increase in anthranilate levels in plasma containing schistosomes vs. control plasma at both time points examined. Anthranilates have been called “endogenous regulators of innate immunity” [59] and have been shown to block both neutrophil infiltration and keratinocyte-derived chemokine expression in a mouse model of psoriatic inflammation [60].

### 4.3. The Purinergic Halo

Intravascular schistosomes have been shown to have the ability to cleave a collection of host cell purinergic signaling molecules around them (such as pro-inflammatory ATP, pro-thrombotic ADP and immunomodulatory NAD) [61,62,63,64]. These, and related, metabolites are collectively known as the worm’s “purinergic halo” [4]. In the present work, we see the impact schistosomes have on additional purinergic metabolites. We report here that levels of adenine, hypoxanthine and xanthine all increase significantly at one or both timepoints examined. At the same time, inosine levels decline significantly. What mediates these changes is unclear. Earlier work established that the schistosome ectoenzyme SmAP can cleave exogenous AMP to generate adenosine [65], and adenosine is well known to be potently anti-inflammatory [66,67]. We previously reported that, in this experimental system, there is an increase in extracellular adenosine in murine plasma over time without a concomitant decrease in AMP levels [64]. This suggests that the action of SmAP does not drive the increase in adenosine observed. Perhaps the parasites release adenosine and this is selectively advantageous because of adenosine’s anti-inflammatory properties? Likewise, adenine may be exported by the worms since its level in plasma increases about 10-fold over the course of our experiment. Like adenosine, adenine too has been reported to exert an anti-inflammatory effect through the activation of adenine receptor (AdeR) signaling in, e.g., mouse macrophages [68]. Similarly, human macrophages treated with adenine before LPS stimulation show reduced inflammatory gene expression, cytokine secretion and surface marker expression [69].

Adult schistosomes incubated in the presence of adenosine are reported to preferentially deaminate it to inosine, which is cleaved to generate hypoxanthine [70], and it appears that this pathway is engaged here too as revealed by the data presented. Inosine is, like adenosine, anti-inflammatory [71]. It can decrease the production of pro-inflammatory mediators in macrophages, neutrophils and lymphocytes, and is protective in animal models of sepsis and autoimmunity [72]. However, here we show that inosine levels decrease naturally in plasma, even in the absence of schistosomes; schistosomes speed up this process, lessening the likelihood that inosine plays a major role in schistosome metabokine-driven immunomodulation.

While the impact of hypoxanthine and xanthine on immune function has been little studied, immunomodulatory effects of xanthine derivatives (like theophylline) have been reported; these actions include inhibition of cytokine synthesis and release, inhibition of inflammatory cell activation, and acceleration of granulocyte apoptosis [73]. So, it is conceivable that extracellular hypoxanthine and/or xanthine impact immune outcomes in schistosomiasis.

### 4.4. Additional Diverse Metabokines

Several additional metabokines belonging to diverse biochemical classes show striking changes in their levels in plasma in the presence of schistosomes. Three of the most dramatic are myo-inositol, betaine, and N-acetylglucosamine. Levels of each of the three increase several-fold in plasma in the presence of schistosomes at 20 min, and further increase by the 60 min time point.

Myo-inositol, a sugar alcohol that is a stereoisomer of cyclohexane-1,2,3,4,5,6-hexol is a component of many structural lipids and secondary messengers [3]. Exogenous myo-inositol exposure has been shown to have anti-inflammatory and antioxidant effects in a human vascular cell model; myo-inositol significantly decreased monocyte–endothelial cell adhesion induced by TNF-α and significantly reduced TNF-α-induced VCAM-1 and ICAM-1 expression [74]. In a murine model of traumatic brain injury, myo-inositol treatment was linked with upregulation of genes in the brain, including those controlling inflammation [75].

Betaine is also known as trimethylglycine. While its primary reported role is to act as a methyl group donor in liver metabolism, betaine also has anti-inflammatory functions, e.g., by acting to inhibit NF-κB and NLRP3 inflammasome activity [76]. Additionally, betaine was shown to depress levels of pro-inflammatory cytokines such as TNF-α, IL-1β, IL-6, and IL-8 in primary human corneal epithelial cells exposed to hyperosmotic stress [77].

N-acetylglucosamine (GlcNAc) is a key monosaccharide involved in glycosylation and the hexosamine biosynthetic pathway. Oral GlcNAc administration inhibits Th1 and Th17 responses and blocks the secretion of IFN-γ, TNF-α, IL-17 and IL-22 in mice [78]. Exogenous GlcNAc can decrease the expression levels of IL-6, IL-8 and TNF-α, which are ordinarily induced following influenza A virus infection of Madin-Darby Canine Kidney (MDCK) cells [79].

Gentisate (2,5-dioxybenzoate) exhibits anti-inflammatory and antioxidant activities [80]. Levels of gentisate increase significantly in plasma containing schistosomes at both time points. Administration of the isoflavone genistein (5,7-dihydroxy-3-(4-hydroxyphenyl)-4H-1-benzopyran-4-one,4′,5,7 trihydroxyisoflavone) can reduce serum levels of pro-inflammatory cytokines like TNF-α and IL-6 in rats [81]. Other anti-inflammatory activities include inhibition of pathways driven by NF-κB, prostaglandins and pro-inflammatory cytokines (reviewed in [82]). Given genistein’s broad anti-inflammatory nature, any selective advantage for schistosomes to decrease the level of this chemical around them, as seen here, is unclear.

There is a marginal, though significant, increase in the level of sorbitol, but only at the 60-minute time point, and there is a similar modest decrease in the level of azelate. Sorbitol, a sugar alcohol, has been reported to enhance the stimulation of C57BL/6 mouse bone marrow-derived dendritic cells (BMDCs) by human papillomavirus-like particles, as revealed by increased levels of marker proteins CD40, CD80, MHC II and CD54 at the cell surface [83]. Azelate is a salt/ester of azelaic acid, and this has also been shown to suppress IL-1β, IL-6 and TNF-α mRNA expression and protein secretion in ultraviolet light-treated human keratinocytes [84].

Fructose exposure ex vivo supports an inflammatory phenotype and promotes elevated cytokine production in LPS-stimulated human or mouse monocytes, and dietary fructose has been shown to increase inflammation in a mouse model [85]. Mean plasma fructose levels do fall in the presence of schistosomes, perhaps because the worms take in the sugar. While the declines noted could lessen any pro-inflammatory impact of exogenous fructose, the values recorded are not significantly different from control.

Over time, the level of glycerol ticks up in plasma without schistosomes but ticks down in the presence of schistosomes. Glycerol can have anti-inflammatory effects [86]; in a mouse model of dermal irritation, glycerol treatment diminished the accumulation of neutrophil granulocytes and lymphocytes, and decreased mRNA levels of the inflammatory cytokines IL-1ß and TNF-α [87].

## 5. Conclusions

Overall, our data show that adult schistosomes by themselves can have a profound impact on the metabolomic profile of murine plasma. We focus on known small molecule immunomodulators and, at both timepoints examined here, we report that in plasma containing adult worms, there are significant increases in the following metabokines: lactate, succinate, 2-hydroxyglutarate, adenine, hypoxanthine, xanthine, myoinositol, betaine and N-acetylglucosamine. As outlined above, each of these compounds can have immunosuppressive effects and, as such, all could impact host immunological status and contribute to schistosome survival.

Of particular interest are changes in levels of metabolites that are known to skew immune responses in a manner that is seen following schistosome infection, such as by promoting Th2 immunity (succinate), Treg generation (lactate) and M2 macrophage polarization (lactate and succinate). However, how relevant these findings are for schistosomes in vivo remains unknown, since it seems likely that metabolites generated around schistosomes in the vasculature would be rapidly diluted by the flow of blood around and past them. On the other hand, a consistent low-level release of selected metabolites might create a local zone of immunosuppressive metabolites immediately around the worms that could impact the biochemistry of any approaching immune effector cells. The pitted nature of the worm’s integument may be relevant here since it could permit the transient accumulation of immunomodulatory metabolites within tegumental pits from where they might gradually diffuse. Of consideration too is the fact that we report on just a small number of compounds from among the several thousand metabolites in plasma.

Levels of some of the metabolites measured here change significantly at just one of the time points examined (e.g., aconitate, sorbitol, azelate) and often the changes are modest (though statistically significantly different from the control). How these may impact the immune environment in vivo is especially hard to discern. It seems likely that some of the changes reported here are biproducts of normal metabolism and have little or no immunomodulatory impact in vivo. Of note, levels of several other known potent metabokines, including many TCA cycle and kynurenine pathway intermediates, do not change to any great degree in plasma containing schistosomes.

Most research work on the impact of metabokines has examined the impact of each one individually, not in combination. Here, we show that schistosomes can generate a cocktail of compounds, and we assume that each one still functions as reported when tested alone, but this may not be the case. Another caveat of this study is that the changes reported are, in each case, given relative to the zero-minute (starting plasma) sample, but they are not quantitative. Therefore, even large changes in relative amounts of a metabolite (e.g., the ~10-fold increase in adenine levels reported here) may amount to more modest changes in the actual quantity of the metabolite.

Studies comparing the metabolomic profiles of individuals infected with schistosomes versus those without schistosomes reveal large differences between the two. Differences include increases in serum/plasma levels of some of the metabolites that are also raised here, such as lactate [88] and succinate [89]. However, while changes described in this work are brought about directly by adult schistosomes in plasma, metabolomic changes in whole infected animals could be linked not just to the adult worms but to, e.g., worm eggs and/or to parasite-induced pathology and/or to changes in gut microflora—all of which could impinge on the metabolome. Our work here shows that the parasites alone could be largely responsible for, or could be contributing to, some of the metabolomic changes reported in infected animals.

How do schistosomes bring about the changes reported here? Are the metabolites that increase in level generated externally or are they released by the worms into the plasma (or both)? Are the metabolites that decrease in the plasma catabolized externally or are they taken up by the worms from the plasma (or both)? If molecules are imported or exported, how exactly do they move in or out of the worms? For some, this is known: lactate and glycerol can be transported across the tegument via aquaporins [23]. In other systems, metabolites such as succinate cross cell membranes largely via monocarboxylate transporter (MCT) proteins [90], but no MCT homologues have been identified in several proteomic studies of the schistosome tegumental membranes. Of course, the worms could release succinate and/or other metabokines into the environment via their protonephridial (excretory) system and not across the tegument. Either way, specific transporters have not been described.

Finally, emphasis here has been placed on the reported ability of the examined metabokines to impact host immunity, and this topic has been much studied. However, it must be selectively advantageous for schistosomes to be able to manipulate other, less well examined, aspects of host physiology and biochemistry, such as vascular tone and host coagulation pathways. Some metabokines are known to impinge on these. For instance, adenine has been reported to be vasoactive [68] and lactate has been shown to inhibit platelet aggregation and impair the coagulation system [91,92]_._ Thus, there are likely many competing pressures on schistosomes and their hosts that drive the rate and amount of metabolite generation and release into the extracellular environment.

## Figures and Tables

**Figure 1 pathogens-14-00526-f001:**
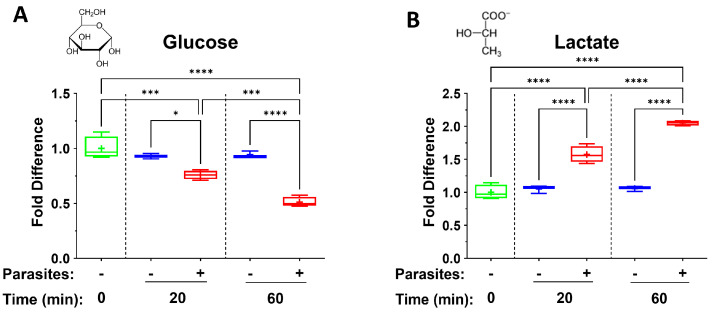
Box plots illustrating changes in levels of glucose (**A**) and lactate (**B**) in plasma containing adult schistosomes (+), or in control plasma lacking schistosomes (−), at the indicated time points. Samples are normalized relative to the 0 min time point (set at 1). The level of glucose significantly decreases over time and the decrease detected at 20 min post incubation becomes even greater at the 60 min time point. In contrast, as shown in (**B**), levels of lactate increase significantly in like manner. The chemical structures of both metabolites are depicted (top left). Each box represents the interquartile range; the horizontal line in each box indicates the median and the “+” denotes the mean. Whiskers represent maximum (upper) and minimum (lower) values. Multigroup statistical analysis was conducted using one way analysis of variance (ANOVA), with significant levels indicated as follows: * *p* < 0.05, *** *p* < 0.001, **** *p* < 0.0001.

**Figure 2 pathogens-14-00526-f002:**
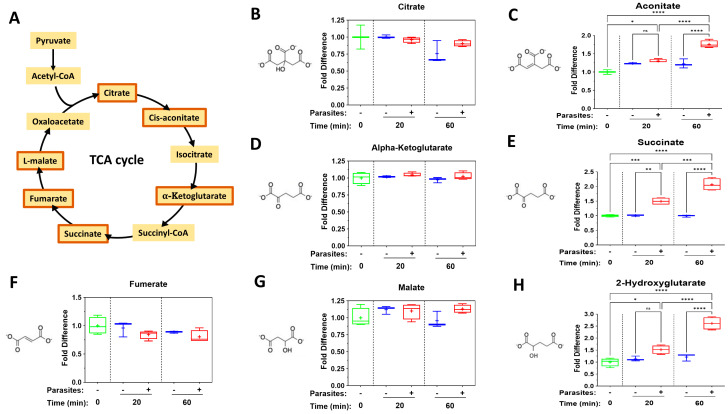
Box plots illustrating changes in levels of tricarboxylic acid (TCA) cycle intermediates measured in plasma containing adult schistosomes (+), or in control plasma lacking schistosomes (−), at the indicated time points. Samples were normalized relative to the 0 min time point (set at 1). (**A**) is a representation of the TCA cycle and bounded in orange boxes are those metabolites measured here. These are citrate (**B**), aconitate (**C**), α-ketoglutarate (**D**), succinate (**E**), fumarate (**F**), malate (**G**) and the α-ketoglutarate derivative, 2-hydroxyglutarate (**H**). The chemical structures of all metabolites are depicted. Significant differences between levels of a metabolite in plasma containing worms vs. equivalent, control plasma lacking worms are indicated. Each box represents the interquartile range; the horizontal line in each box indicates the median and the “+” denotes the mean value. Whiskers represent maximum (upper) and minimum (lower) values. Multigroup statistical analysis was conducted using one way analysis of variance (ANOVA), with significant differences indicated as follows: * *p* < 0.05, ** *p* < 0.01, *** *p* < 0.001, **** *p* < 0.0001. ns, not significant.

**Figure 3 pathogens-14-00526-f003:**
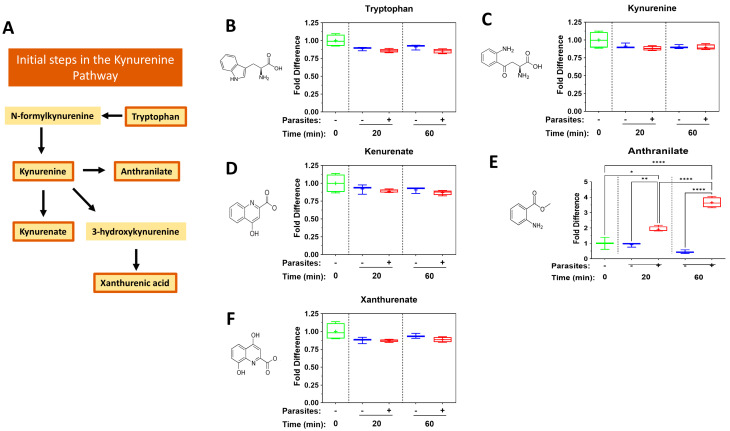
Box plots illustrating changes in levels of metabolites of the kynurenine pathway measured in plasma containing adult schistosomes (+) or in control plasma lacking schistosomes (−) at the indicated time points. Samples were normalized relative to the 0 min time point (set at 1). (**A**) is a representation of part of the kynurenine pathway; bounded in orange are those metabolites measured here: tryptophan (**B**), kynurenine (**C**), kynurenate (**D**), anthranilate (**E**) and xanthurenate (**F**). The chemical structures of all metabolites are depicted. Significant differences between levels of a metabolite in plasma containing worms vs. equivalent, control plasma lacking worms are indicated. Each box represents the interquartile range; the horizontal line in each box indicates the median and the “+” denotes the mean value. Whiskers represent maximum (upper) and minimum (lower) values. Multigroup statistical analysis was conducted using one way analysis of variance (ANOVA), with significant levels indicated as follows: * *p* < 0.05, ** *p* < 0.01, **** *p* < 0.0001.

**Figure 4 pathogens-14-00526-f004:**
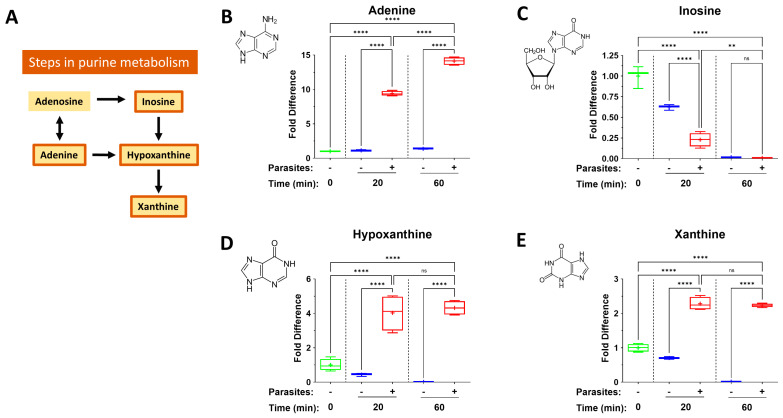
Box plots illustrating changes in levels of purinergic metabolites measured in plasma containing adult schistosomes (+), or in control plasma lacking schistosomes (−), at the indicated time points. Samples were normalized relative to the 0 min time point (set at 1). (**A**) illustrates some steps in purine metabolism; the remaining panels depict levels of adenine (**B**), inosine (**C**), hypoxanthine (**D**) and xanthine (**E**) over time. The chemical structures of all metabolites are depicted. Significant differences between levels of a metabolite in plasma containing worms vs. equivalent, control plasma lacking worms are indicated. Each box represents the interquartile range; the horizontal line in each box indicates the median and the “+” denotes the mean value. Whiskers represent maximum (upper) and minimum (lower) values. Multigroup statistical analysis was conducted using one way analysis of variance (ANOVA), with significant levels indicated as follows: ** *p* < 0.01, **** *p* < 0.0001. ns, not significant.

**Figure 5 pathogens-14-00526-f005:**
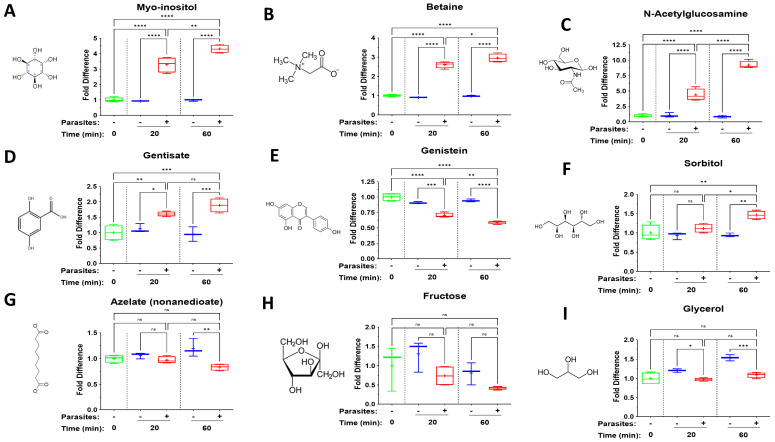
Box plots illustrating changes in levels of the following metabolites in plasma containing adult schistosomes (+), or in control plasma lacking schistosomes (−), at the indicated time points: myo-inositol (**A**), betaine (**B**), N-acetylglucosamine (**C**), gentisate (**D**), genistein (**E**), sorbitol (**F**), azelate (**G**), fructose (**H**) and glycerol (**I**) over time. Samples were normalized relative to the 0 min time point (set at 1). The chemical structures of all metabolites are depicted. Significant differences between levels of a metabolite in plasma containing worms vs. equivalent, control plasma lacking worms are indicated. Each box represents the interquartile range; the horizontal line in each box indicates the median and the “+” denotes the mean value. Whiskers represent maximum (upper) and minimum (lower) values. Multigroup statistical analysis was conducted using one way analysis of variance (ANOVA), with significant levels indicated as follows: * *p* < 0.05, ** *p* < 0.01, *** *p* < 0.001, **** *p* < 0.0001. ns, not significant.

**Table 1 pathogens-14-00526-t001:** List of 16 murine plasma metabokines (of 27 examined) whose levels are significantly altered in the presence of adult *S. mansoni* parasites at 20 and/or 60 min. Data are shown relative to levels in control plasma, lacking worms (mean fold difference). Metabokines are arranged in descending order, based on fold change at 60 min. Gray boxes represent the three metabolites whose levels decreased significantly in the presence of the worms.

Metabokine	Mean Fold Change at 20 min (+Parasites)	Mean Fold Change at 60 min (+Parasites)
Adenine	9.4	14.1
N-Acetylglucosamine	4.4	9.2
Hypoxanthine	4.1	4.3
Myo-inositol	3.3	4.3
Anthranilate	1.9	3.7
Betaine	2.6	3.0
2-Hydroxyglutarate	1.5	2.6
Xanthine	2.3	2.2
Succinate	1.5	2.1
Lactate	1.6	2.1
Gentisate	1.6	1.9
Aconitate	1.3	1.8
Sorbitol	1.1	1.5
Genistein	0.7	0.6
Glucose	0.8	0.5
Inosine	0.2	0.01

## Data Availability

The original contributions presented in this study are included in the article/Appendix A. Further inquiries can be directed to the corresponding author.

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
