# Peer review of "Generation of Immune Modulating Small Metabolites—Metabokines—By Adult Schistosomes"

_pathogens, 2025, doi:10.3390/pathogens14060526_

Round 1
Reviewer 1 Report
Comments and Suggestions for Authors
Dear authors,
Congratulations for the manuscript (MS). I hope my suggestions may improve the content and make the MS better for the journal and the readers.
My major issue resembles the raw data of the metabolomic analysis. Table 1, which should be presented in the Results section, may present the raw data and not just the mean fold change of the significantly altered metabolites. Another possibility is creating a Supplementary table comprising the raw data of the study. There is no doubt about the results, but reporting the data the study becomes clearer, without a doubt, transparent.
Minor questions which should improve the MS are:
- There are so many reviews. More credit could be given to authors from experimental investigations;
- “The relative levels of selected metabokines are described below” in line 98 is not a Methodology description. Please, consider removing or sending it to the Results section;
- The paragraph on lines 137 to 137 should be inserted before Figure 1. The figure fits better after the description of the result;
- Please, replace “too” for “also” in line 316 of the Discussion section;
- The subsections division on the Discussion section is not necessary;
- Table 1 should be presented in the Results section, not on Conclusion;
- The paragraph on lines 502 to 514 provides a discussion of the results, not a conclusion. Please, consider inserting this paragraph between your discussion;
- The three paragraphs between line 525 to 556 are not necessary. There is so much information and data already discussed, and it´s not a conclusion.
- As previously reported, the raw data may be provided. Therefore, the declaration on line 577 “Data Availability Statement: Not applicable—all data presented herein” is not true…
Author Response
First, we thank the reviewer for the kind congratulations.
Reviewer comment: My major issue resembles the raw data of the metabolomic analysis. Table 1, which should be presented in the Results section, may present the raw data and not just the mean fold change of the significantly altered metabolites. Another possibility is creating a Supplementary table comprising the raw data of the study. There is no doubt about the results, but reporting the data the study becomes clearer, without a doubt, transparent.
Response: As suggested, we have moved the Table into the Results section and amended the text to reflect this (lines 232 – 239). In addition, as recommended by the reviewer, we have created a Supplementary Table (S1) which contains the raw data. We note this in the revised text on lines 103 – 105 and on line 233.
Reviewer comment: Minor questions which should improve the MS are:
There are so many reviews. More credit could be given to authors from experimental investigations;
Response: While we have, as recommended, added additional references in the revised manuscript crediting individual experimental groups (e.g., new references 18, 71 and 86 on lines 163, 403 and 465), we also elect to retain references to several review papers, since these will aid readers who wish to learn more.
Reviewer comment: “The relative levels of selected metabokines are described below” in line 98 is not a Methodology description. Please, consider removing or sending it to the Results section;
Response: We have removed the text on (old) line 98, as suggested by the reviewer.
Reviewer comment: The paragraph on lines 137 to 137 should be inserted before Figure 1. The figure fits better after the description of the result;
Response: We thank the reviewer for the suggestion and have moved the paragraph, as suggested. (Now, lines 122 – 125).
Reviewer comment: Please, replace “too” for “also” in line 316 of the Discussion section;
Response: We have replaced “too” for “also”, as suggested (now line 335)
Reviewer comment: The subsections division on the Discussion section is not necessary;
Response: While the subsections delineated in the Discussion are not strictly necessary, we do think that they provide useful boundaries for readers and therefore we elect to retain them in the revised work.
Reviewer comment: Table 1 should be presented in the Results section, not on Conclusion;
Response: As noted above, Table 1 is now inserted in Results, as the reviewer requests.
Reviewer comment: The paragraph on lines 502 to 514 provides a discussion of the results, not a conclusion. Please, consider inserting this paragraph between your discussion; The three paragraphs between line 525 to 556 are not necessary. There is so much information and data already discussed, and it´s not a conclusion.
Response: While the reviewer is correct that the paragraph in question is not strictly a “conclusion”, we feel that it aids the flow of the paper to retain it where it is. Similarly, we feel that the paragraphs mentioned contain information that readers will find helpful, and therefore we retain that text in the updated manuscript. These are not major points and if the editor feels that the requested changes are essential, we can comply.
Reviewer comment: As previously reported, the raw data may be provided. Therefore, the declaration on line 577 “Data Availability Statement: Not applicable—all data presented herein” is not true…
Response: As mentioned earlier, the raw data are now included as Supplementary Table S1 and the declaration on (now) line 559 is correct.
Reviewer 2 Report
Comments and Suggestions for Authors
The manuscript explores changes in small metabolite levels in plasma occupied by adult schistosomes. The idea is quite novel, and the methodology appears appropriate. The results are descriptive and presented clearly.
(1) I wonder whether the term "immune modulating" in the manuscript title might be overstated. While small metabolites can act as immunomodulators under certain circumstances, this study does not demonstrate such effects in the context of schistosomiasis. I suggest adding the word "potential" to reflect that the immunomodulatory effects have not been directly verified.
(2) How long were the parasites cultured in vitro (lines #83–86)? I also could not find the results corresponding to this part of the experiment. Apologies if I missed it, but it seems that the presented results originate only from the incubation in plasma, is that correct?
(3) It is unclear when a t-test was used. The figure legends state that only ANOVA (presumably with Tukey's post hoc test) was applied, which seems appropriate and sufficient for all indicated comparisons. Therefore, the t-test may be unnecessary.
(4) The Discussion and Conclusions place the descriptive findings into a broader context and propose possible interactions and effects. While this is appreciated, the section is quite long - in fact, considerably longer than the Results section.
Author Response
Reviewer comment: (1) I wonder whether the term "immune modulating" in the manuscript title might be overstated. While small metabolites can act as immunomodulators under certain circumstances, this study does not demonstrate such effects in the context of schistosomiasis. I suggest adding the word "potential" to reflect that the immunomodulatory effects have not been directly verified.
Response: While we see the reviewer’s point, our data do show that adult schistosomes can alter the levels of many metabokines. Since several of these are clear, undisputed immunomodulators, we feel that the current title is valid. In addition, we argue that we are sufficiently circumspect in the manuscript regarding the relevance of the changes in the context of schistosome infection (e.g., lines 481 – 489) and one aim of the work is to promote further study on this point.
Reviewer comment: (2) How long were the parasites cultured in vitro (lines #83–86)? I also could not find the results corresponding to this part of the experiment. Apologies if I missed it, but it seems that the presented results originate only from the incubation in plasma, is that correct?
Response: The reviewer is correct; the times mentioned in the paper refer to time incubated in plasma. The worms had previously been in culture for 48 hours and this information is now presented in the Methods section (line 83).
Reviewer comment: (3) It is unclear when a t-test was used. The figure legends state that only ANOVA (presumably with Tukey's post hoc test) was applied, which seems appropriate and sufficient for all indicated comparisons. Therefore, the t-test may be unnecessary.
Response: Again, the reviewer is correct, and we have removed mention of t-tests in the revised work.
Reviewer comment: (4) The Discussion and Conclusions place the descriptive findings into a broader context and propose possible interactions and effects. While this is appreciated, the section is quite long - in fact, considerably longer than the Results section.
Response: Moving the Table and accompanying text from the Discussion to the Results section, as suggested by reviewer 1, has lessened the length of the revised Discussion. That said, recall that we have examined changes in the levels of 27 compounds and we feel that it is important to discuss each in turn. We report on what is known about the immunomodulatory role of each compound as well as its potential importance in the context of schistosome infection and these factors contribute to a (necessarily) lengthy Discussion section.
Reviewer 3 Report
Comments and Suggestions for Authors
General comments
In this article, the Authors report the research findings on a set of biochemical molecules (metabokines), and their potential role to modulate the immune host response induced by S. mansoni. I indeed agree that the topic is of great importance, as many immune evasion strategies used by these parasites (and others), still remain intriguing and unclear. This is a very well written manuscript and despite the complexity of the experiments, results are clear, well presented and justified.
Minor point: line 77 – italic Schistosoma mansoni
Author Response
We thank the reviewer sincerely for the positive overview.
Reviewer comment: Minor point: line 77 – italic Schistosoma mansoni
Response: We have italicized Schistosoma mansoni, as recommended (line 77).